# Challenges and Opportunities for Software Testing in Virtual Reality Application Development

Qing Liu*
School of Computer Science
University of Waterloo

Gustavo Alves †
Unity Technologies

Jian Zhao‡
School of Computer Science
University of Waterloo

## ABSTRACT

Testing is a core process for the development of Virtual Reality (VR) software, which could ensure the delivery of high-quality VR products and experiences. As VR applications have become more popular in different fields, more challenges and difficulties have been raised during the testing phase. However, few studies have explored the challenges of software testing in VR development in detail. This paper aims to fill in the gap through a qualitative interview study composed of 14 professional VR developers and a survey study with 33 additional participants. As a result, we derived 10 key challenges that are often confronted by VR developers during software testing. Our study also sheds light on potential design directions for VR development tools based on the identified challenges and needs of the VR developers to alleviate existing issues in testing.

## 1 INTRODUCTION

Debugging or testing is one of the critical steps in software development [29]. The creation of Virtual Reality (VR) applications shares a similar process to traditional software development and heavily relies on testing to ensure the quality of the final deliverables. However, VR application testing is more challenging and complex [2, 27] due to its inherent nature of relying on multiple devices and platforms including headsets and desktops. For instance, developers need to put on and take off their VR head-mounted display (HMD) quite frequently during the testing stage, which not only is time-consuming but also causes motion sickness [20]. In addition, developers do not always have access to VR HMDs to realistically evaluate the quality of their creations.

While the human-computer interaction (HCI) community is increasingly focused on researching immersive technologies such as VR and augmented reality (AR), there is still a lack of thorough studies exploring the challenges as well as opportunities for software testing in VR development. Various studies have explored AR/VR applications (e.g., [21, 26, 37]), interaction techniques (e.g., [18, 25, 32, 33]), and authoring tools (e.g., [4, 7, 19]). Some have provided insights into the challenges and opportunities of AR/VR from the development perspective to better understand the needs of professional AR/VR developers [2, 11, 14, 22, 27]. However, little attention has been paid to the challenges of the testing phase in VR development, despite developers having confronted numerous difficulties as introduced above.

In this research, we aimed to fill in the gap by exploring the challenges and needs of VR developers during their testing phase and identifying promising directions to overcome or resolve these challenges. We first conducted a comprehensive interview study with 14 professional VR developers (11 from industry and 3 from academia) who have diverse backgrounds and different levels of experience

---

*e-mail: qing.liu@uwaterloo.ca
†e-mail: gustavo.alves@unity3d.com
‡e-mail: jianzhao@uwaterloo.ca

and skill sets in VR development. We then performed a thematic analysis of the interview data and identified 10 key challenges for software testing in VR development. Our results confirmed that VR developers face significant challenges during the testing phase of VR development. Despite employing workarounds, our participants found them to be ad-hoc, requiring manual intervention, and prone to errors. We organized the key challenges into three distinct categories (see Table 2): hardware-related challenges (C1-4), software-related challenges (C5-8), and comprehensive challenges (C9-10).

To verify the identified challenges with a broader audience, we further conducted a confirmation survey with 33 VR developers by distributing the survey to various related Slack channels. In the survey, we asked participants to rank the importance of the 10 challenges on a 7-point Likert scale as well as select the most and least important ones. From the survey results, all the identified challenges exhibit reasonable ratings without any outliers, substantiating the validity of our findings.

Additionally, we discuss the future opportunities for testing VR applications based on the identified challenges, which can provide guidance to VR developer tool makers and researchers for enhancing the current functionalities of VR development tools and introducing new features. Our study extends the findings of Ashtari et al. [2], Nebeling and Speicher [27], and Krauß et al. [22], which benefits specifically to the testing phase of VR development. In summary, our contributions in this paper include:

- Empirical interview and survey studies that examined and validated key challenges in the testing phase of VR development;
- The results of 10 key challenges in VR testing faced by developers as well as several promising future design directions.

## 2 RELATED WORK

Our research is related to the existing techniques of VR development and authoring tools, practices in VR development, as well as studies on VR testing and general software testing.

### 2.1 Development and Authoring Tools for VR

Development tools for VR assist creators with varying expertise levels in producing VR software. VR development tools encompass 3D game engines (e.g., Unity, Unreal, Godot) and development toolkits/frameworks (e.g., MRTK, A-Frame). These tools empower VR developers and researchers to create versatile VR applications.

Based on these tools, several research studies [4, 6, 10, 13, 19, 39] have proposed VR authoring tools to satisfy the customized needs of the end-users. These tools focus on satisfying some specialized needs of the end users. Some systems (e.g., *VREX* [4], *Xr360* [19], *Genesys* [10]) are designed to lower the threshold of VR development and speed up the process. Some other studies help the users meet specialized needs such as creating interactive scenes [39], making experiential learning courses [6], and authoring VR games for physical space [13].

An existing study [9] shows that VR authoring tools could help facilitate the creation of different VR features. Despite the availability of the existing authoring tools, there has been a limited number of tools created and researched to ease the VR testing practices for developers. Additionally, with the recent explosion in VR systems

and applications in different fields (e.g., [21, 26, 37]), it is important to understand the needs of VR developers during the development process, which has motivated our study. Our study provides empirical insights into the testing of VR development, highlighting the challenges faced by VR developers with some future directions for the creation of new tools that could facilitate VR testing.

## 2.2 VR Development Practices

To better understand the challenges and needs in the testing phase of VR development, our study aims to investigate the current development practices of VR developers. Unity and Unreal engines are the common development tools used by VR developers. An explanatory study by Ghrairi et al. [15] discovered that the majority of VR projects on GitHub are currently small to medium-sized, with JavaScript (used for web) and C# (used in Unity) being the most popular programming languages. Unity has emerged as the preferred game engine for VR development and is the most frequently discussed topic on Stack Overflow. In addition to Unity, Unreal Engine is also utilized by VR developers and researchers for creating VR content [5, 8] Unity is preferred for its ease of use, asset store, and support for various platforms, while Unreal Engine is favoured for its advanced graphics capabilities and visual scripting [34]. Numerous customized tools, such as Unity XR Interaction Toolkit and MRTK, have been developed to streamline and support the VR development process in Unity or Unreal Engine.

On the other hand, the field of web-based VR development, particularly through the implementation of WebXR, has experienced significant growth in recent years. WebXR, an API that enables the creation and integration of immersive experiences directly within web browsers, has emerged as a popular alternative for the industry. By allowing developers to create platform-agnostic VR experiences [23], WebXR fosters accessibility and reduces the need for specialized hardware or software. As a result, researchers are increasingly exploring the potential of web-based VR development for a wide range of applications, such as education [16, 26, 28], healthcare [1], and entertainment [17]. The growing interest in WebXR also highlights the importance of developing new tools, frameworks, and best practices to support the unique challenges and opportunities associated with web-based VR development [40].

Cross-platform VR development has also been promoted these years by organizations like Khronos Group and its open standard OpenXR. However, significant challenges still remain for developers. They face a multitude of issues when working with different VR hardware, software, and application programming interfaces (APIs). Heterogeneous specifications, input mechanisms, and performance capabilities can lead to compatibility and optimization difficulties, requiring developers to adapt their applications to each unique platform. Additionally, the varying degrees of support for industry standards and the rapid evolution of VR technology further complicate cross-platform development. VR development practices have made considerable advancements, thanks to the widespread adoption of game engines like Unity and Unreal, the emergence of web-based VR development through WebXR, and the push for cross-platform development by organizations like Khronos Group. These improvements have resulted in more efficient development processes and the creation of customized tools and frameworks. However, challenges persist in the realm of VR development, including compatibility and optimization issues across diverse hardware, software, and APIs, as well as the rapid evolution of VR technology and varying support for industry standards [2, 11, 22]. To ensure the continued growth and success of the VR industry, it is essential to address these challenges, foster collaboration among developers, and researchers, and continue exploring new methods and best practices to improve the VR development process and enhance the user experience, where our study aims to contribute to the understanding the specific challenges in the testing phase of VR development process.

## 2.3 Testing Practices in Software Development

Software testing, which is an essential step in the whole software development workflow [12], has been researched to ensure the quality of the software delivery [3, 35, 38]. While testing in VR software development has not been studied comprehensively, general software testing could still enlighten the directions of VR software testing in different ways.

Automated testing, as an important practice of software testing, has been implemented and applied to the software industry broadly. Automated testing has significantly impacted the testing process, with many software tests now being performed using automation tools [35]. These tools reduce the number of people involved and the likelihood of human errors. Automated testing involves test cases that simplify the process of capturing different scenarios and storing them. For example, automated tests have been explored to reduce the errors in software GUI [24].

Manual testing is also an important practice of software testing and has been researched in comparison with automated testing [30, 31, 36]. Manual testing always involves human efforts from testing teams such as Quality Assurance testers and Software Developers who are responsible for creating and running tests. Manual testing is a time-consuming process that demands specific qualities in a tester, such as patience, observance, creativity, open-mindedness, and skill [31]. When applied to large software applications or those with extensive datasets, repetitive manual testing can become challenging to execute effectively. This limitation underscores the need for alternative methods, such as automated testing, to improve efficiency and accuracy in software testing processes. However, due to the nature of VR applications, manual testing is still inevitable as VR software relies on human work to ensure the quality of the products such as the visual presentation of the contents and graphics performance in the VR headsets.

While the above studies have explored the needs in general software testing and have proposed tools to address the issues, VR testing can be particularly challenging because of its unique development environment with the HMDs. Different software testing techniques might be customized and applied to VR testing to ensure the delivery of VR software; however, no studies have adequately investigated scenarios of VR testing. Thus, our research specifically aims to get insights into the challenges and opportunities in the testing phase of VR development.

## 3 INTERVIEW STUDY

To investigate the current practices and challenges in VR application testing faced by developers in-depth, we employed a qualitative approach by conducting semi-structured interviews with VR developers with diverse backgrounds. In this section, we describe the setup of the interview study and report the results in the next section.

### 3.1 Participants

In order to gain a comprehensive understanding of VR development testing practices, we sought out participants with experience in the field, from both academia and industry. We reached out to local HCI research groups as well as VR-related software companies. Our goal was to create a diverse cohort of participants, with varying backgrounds and project experience. We ultimately recruited 14 participants (11 males, 2 females, and 1 non-binary/third gender; aged 19–54), including user experience designers, gaming enthusiasts, and academic researchers, as detailed in Table 1. Their experience ranged from 0–2 years to 10+ years; and the cohort covered a variety of popular VR hardware (HMD) on the market, including Oculus Quest 1/2, Oculus Rift, HTC Vive, Meta Quest Pro, etc. In addition, our participants use various VR development software (e.g., Unity, Unreal, and Godot) for their work. Based on their experiences and roles in VR development, we grouped them as junior developers (JD), experienced developers (ED), and VR development

Table 1: Participants recruited in our interview study.

| ID | Role | Experience | Software Used | Hardware (HMD) Used |
|----|------|------------|---------------|---------------------|
| **Junior Developers (JD)** | | | | |
| P1 | Software Developer | 0 - 2 years | Godot | Oculus Quest 1/2 |
| P2 | Student Researcher | 0 - 2 years | Unity | Oculus Quest 1/2 |
| P7 | Software Developer | 0 - 2 years | Unity | Oculus Quest 1/2, Oculus Rift, HTC Vive, Google Cardboard |
| P10 | Product Designer | 0 - 2 years | Unity | Oculus Quest 1/2 |
| **Experienced Developers (ED)** | | | | |
| P3 | Student Researcher | 6 - 10 years | Unity | Oculus Rift, HTC Vive |
| P4 | Architectural Designer | 3 - 5 years | Unity, Unreal | Oculus Quest 1/2, Oculus Rift |
| P5 | Software Developer | 3 - 5 years | Unity | Oculus Quest 1/2, Oculus Rift, HP Reverb, Meta Quest Pro |
| P6 | Software Developer | 3 - 5 years | Unity, Unreal | Oculus Quest 1/2, Oculus Rift, HTC Vive, Varjo VR1/2/3 |
| P8 | Software Developer | 10 + years | Unity | Oculus Quest 1/2, Oculus Rift, HTC Vive, Google Cardboard, Valve Index, HP Reverb, Pico, Focus 3, and other Windows enterprise headsets |
| P9 | Software Development Manager | 3 - 5 years | Unity | Oculus Rift, HTC Vive |
| **VR Development Tools Developers (VDTD)** | | | | |
| P11 | Software Development Manager, XR Foundation | 3 - 5 years | Unity, Unreal, Self-build engine | Oculus Quest 1/2, Google Cardboard, HP Reverb, Meta Quest Pro |
| P12 | VR Development Tools Designer | 10 + years | Unity | Oculus Quest 1/2, Oculus Rift, HTC Vive, Google Cardboard |
| P13 | VR Development Tools Developer | 6 - 10 years | Unity, Unreal | Oculus Quest 1/2, Oculus Rift, HTC Vive |
| P14 | VR Development Tools Developer | 3 - 5 years | Unity | Oculus Quest 1/2, Oculus Rift, HTC Vive, Valve Index |

tools developers (VDTD). The diversity in these aspects could provide valuable insights into the testing phase of VR development on different perspectives.

### 3.2 Interview Procedure

All the interviews were conducted remotely via Zoom. Prior to the interview, participants were asked to sign the consent form and filled in a pre-study questionnaire regarding their demographic information. During the interview, we began by asking participants to describe their current or recent VR projects and let them walk through the VR development workflow on the projects they discussed. We then inquired about the testing techniques they used and the main challenges or frustrations of their current testing and debugging process. Our questions were around the following themes during the interviews:

1. Would you briefly introduce one of the interesting VR experiences you had?
2. Could you walk through your VR development workflow on the project you've talked about or another specific example with us?
3. In the walk-through you just shared with us, what were the testing techniques you use?
4. What are the main challenges or frustrations about your current testing workflow?
5. What are your current solutions for the challenges you just mentioned?
6. What could be the ideal VR testing workflow in your mind? It could be a whole workflow, a new tool or some features.

In the end, participants were asked to brainstorm the future directions of VR development tools that could better serve the testing purpose of VR development. The whole interview session was audio recorded and lasted around 60 minutes for each participant. This interview study was approved by the University of Waterloo Research Ethics Board.

### 3.3 Data Analysis

We transcribed the audio recordings of the interview sessions using Otter.ai and manually checked the places that might not be precise due to the limitation of the transcription software. We employed an inductive approach and generated affinity diagrams in Figma to explore the themes related to the main challenges that our participants faced. Initially, one member of our research team conducted an open-coding pass to generate a list of potential codes. We then refined and consolidated these codes through discussions and the use of affinity diagrams, resulting in a final coding scheme. Throughout the coding process, we focused on understanding the challenges and needs of VR developers in their testing practices of VR development.

### 4 CHALLENGES IN VR TESTING

Based on our analysis of the interview data, we consolidated the following 10 key challenges in testing VR applications which are grouped into three categories (Table 2).

### 4.1 Hardware-related Challenges

Hardware-related challenges often arise during the testing phase, posing significant obstacles for developers. These challenges include cumbersome VR equipment (C1), motion sickness (C2), difficult equipment setup (C3), and performance issues (C4). Addressing these hardware-related challenges is essential for streamlining the testing process and ensuring the successful development of VR applications.

**C1: Cumbersome VR Equipment.** Cumbersome VR headsets are a burden for the developers in the testing phase. First, they may suffer from frequently putting on and taking off the headsets, which is not only time-wasting but also triggers feelings of unease or sickness:

> *When I'm using the headset, I have to like, put it on, and then, do stuff, and then put it off, put it away and look at my console, so on and so forth. If there is a perfect simulator I can use to*

Table 2: Summary of challenges in the testing phase of VR development.

| | | |
|---|---|---|
| **Hardware-related Challenges** | | |
| C1 | **Cumbersome VR Equipment** | Developers may suffer from the inconvenience of the VR headsets. For example, developers may experience taking on and off the VR headsets in a high frequency, the sickness caused by the heavy weight of the headsets, or the burden of the eyeglasses and long hair. |
| C2 | **Motion Sickness** | Developers may suffer from motion sickness caused by the VR environment and equipment. The motion sickness might be caused by, for instance, the long time spent in VR environments or the low quality (low frame rate, low picture quality) of VR application during the prototyping/testing phase |
| C3 | **Difficult Equipment Set Up** | Developers may suffer from the difficult and time-wasting equipment set-ups during the testing phase. For example, developers need to recalibrate the VR equipment every time when they use it. It also has a strict demand on an open, decent-size and obstacle-free physical space when developers want to do some trials in VR environments. |
| C4 | **Performance Issues** | Developers may suffer from performance issues during the testing phase. Issues such as long build/loading/rendering time, the discrepancy between hardware's performance (in most cases, the simulator environments like the PC and laptop have better hardware performance than the VR equipment and low frame rate. |
| **Software-related Challenges** | | |
| C5 | **Missing Testing Information** | Developers may suffer from the lack of testing information. Many developers reported that they cannot monitor program changes (variables, hardware usage) in VR environments and it is hard to integrate debug information (e.g. logs) for VR applications. |
| C6 | **Difficulty in Finding/Reproducing bugs** | Developers may find it hard to find or reproduce bugs. For example, the large 3D immersive environment of VR makes it hard to find details/small glitches. It is also difficult to reproduce bugs since it's hard to track and reproduce the same actions in VR environments. |
| C7 | **Lack of Automated Testing** | Developers may suffer from the inconvenience of immature automated testing support. The lack of automated/unit tests in VR makes it hard to reduce manual/repetitive testing work. |
| C8 | **Inconvenient Collaboration of VR Testing** | Developers may find it hard to do collaborative debugging/testing with other developers. For example, developers may find it hard to achieve remote testing/debugging and headset sharing with other developers. |
| **Comprehensive Challenges** | | |
| C9 | **Lack of Standards** | Developers may suffer from the capability issues. Many developers find there are no common standards (e.g. different APIs) between different VR development tools and software (e.g. Unity, Unreal), hardware (e.g VR HMD like Oculus Quest and HTC Vive). |
| C10 | **Few VR-specific Testing Support** | Developers may suffer from the little VR-specific testing support from the community and industry. There are issues such as low-number of existing toolkits, tutorials, documentation or no collaboration/integration between different tools/solutions. |

*test most of the features, it will definitely make debugging a lot easier for me. (P2-JD)*

*From time to time, I got a headache after putting on and off the VR headset to figure out some tricky bugs. (P5-ED)*

Additionally, with the cumbersome HMDs, wearing eyeglasses or having long hair can add to the difficulty experienced during use:

*Having long hair just makes it harder to put on and off the VR headset. (P1-JD)*

*So I don't buy glasses that are wider than that. So that limits my frame choices. (P8-ED)*

One of the VR tool developers expressed concern that this issue might not be resolved in the near future:

*The equipment being cumbersome is like, the sort of issues for which the industry does not have a solution yet, and it might take some time until we find one. (P13-VDTD)*

**C2: Motion Sickness.** Motion sickness caused by VR environments and equipment has been mentioned in a high frequency in the interviews (10/14). This discomfort could be a result of spending extended periods of time in VR environments and encountering low-quality VR applications during the prototyping/testing phase, which may exhibit poor picture quality or low frame rates:

*I used to have a really bad sketch of my project and the whole horizon in VR was shaking with a very low frame rate, and it caused huge dizziness. (P2-JD)*

The issue of cumbersome VR Equipment also further exacerbates the situation:

*If you have to wear glasses, put them on. And you know, it's already like a burden for you. And if you like, do the very frequently, you will have a lot of like headaches and motion sickness. (P7-JD)*

Some VR applications with specific features such as frequent locomotion also contribute to motion sickness:

*One thing to notice is about the locomotion in VR: a lot of them give you motion sickness. (P3-ED)*

Motion sickness could heavily postpone the testing progress of the VR development and may cause the production delay:

*It was difficult for first-time users, I need to adjust things slower for them. (P5-ED)*

*But if your participants or even the developer, start experiencing physical discomfort due to motion sickness, you won't probably be able to have a sustained session on the headset, and therefore, it really impacts what you can get out of the testing. (P12-VDTD)*

**C3: Difficult Equipment Set Up.** Developers may struggle with time-consuming and challenging equipment setups during the testing phase. For instance, they are required to recalibrate or even reboot the VR equipment because the existing calibration was easy to break.

*I need to recalibrate my headsets a lot of time during the testing, sometimes even rebooting the machine, because sometimes the responding calibration was okay, and then the next time the calibration was not good anymore. We have to recalibrate and install things. (P6-ED)*

Additionally, there is a stringent need for a spacious, open, and obstruction-free physical area when developers wish to conduct trials within VR environments:

*I need to clean up the physical space around me every time before some intensive VR testing, being able to fake a physical system in which you don't need to really move to test would save my time a lot. (P3-ED)*

**C4: Performance Issues.** During the testing phase, developers may encounter performance-related challenges. Such challenges can involve prolonged build, loading, and rendering times, leading many participants to invest additional time in testing their VR projects.

Long build, loading or rendering time extended the testing phase of VR development:

*When I checked in many changes to a big VR project, more than half of the hour was waiting for the build. (P5-ED)*

*Rebooting the machine took a lot of time since it's not only rebooting the machine itself but sometimes you have to rebuild and reload the project. (P6-ED)*

The disparities in the performance of hardware components also pose difficulty to the testing. Typically, simulator environments such as PCs and laptops have superior performance compared to VR equipment. As a result, simulators cannot substitute VR equipment when it comes to testing the performance of VR projects.

*When testing performance, the simulator gives you nothing, you have to build on the VR headset to know if the app runs well (P5-ED)*

*We had a project on Oculus Quest...because it is an Android app and has a big resolution, we needed to cut many features to accommodate the performance limitation. (P5)*

*It is hard to check the performance issue without running it on headsets. (P6-ED)*

The low frame rate, which has been raised by 6 participants, can also be caused by low performance, which would cause motion sickness (C2) as discussed before and uncertainty to the projects:

*If the frame rate sucks, motion sickness would probably come. (P7-JD)*

*The different frame rates give different glitches all the time. (P6-ED)*

### 4.2 Software-related Challenges

Other than hardware-related challenges, developers often face a variety of software-related challenges that impact the testing phase of VR development. These challenges include a lack of testing information (C5), difficulty in finding and reproducing bugs (C6), lack of automated testing (C7), and inconvenient collaboration of VR testing (C8). From software developers' perspective, software-related challenges are relatively easier to mitigate. Addressing these software-related challenges is crucial for optimizing the VR development process and ensuring the creation of high-quality applications.

**C5: Missing Testing Information.** Developers might face difficulties due to a lack of adequate testing information. Many developers, especially junior developers, have indicated that tracking program changes, including variables and hardware usage, within VR environments poses a challenge, and incorporating debug information, such as logs, into VR applications also proves to be problematic.

*Some of this can be really frustrating, because, it's sometimes very hard to see, like the internal state of the system, which you kind of need for debugging. (P2-JD)*

*Ideally, if there is some log, or debugging options to track those variables, that would be ideal. (P1-JD)*

Furthermore, some more experienced participants suggested that VR development tools could even do more than just show the basic testing information. For example, for Unity developers, an integrated Unity inspector in the VR environment has been raised as a missing component for VR testing:

*I would love to see in headset authoring. So be able to like, you know, put on the headset, and basically be able to see the scene hierarchy and have control over your inspector at least for some parts, you know, basically like an engine within the headset. (P12-VDTD)*

**C6: Difficulty in Finding/Reproducing bugs.** Developers can face challenges when trying to identify or reproduce issues in VR applications. The vast 3D immersive environment can make pinpointing minute details or minor inconsistencies difficult:

*Sometimes, I needed to go back frame by frame to check the bug I saw. (P1-JD)*

*It's tedious to reproduce bugs. It may not be actually challenging It's just umm, you just need to take time to reproduce it. (P3-ED)*

Additionally, recreating bugs can prove to be troublesome, as retracing and replicating the precise actions within VR settings can be a complex task compared to reproducing them in the simulator:

*You might be testing your experience in the editor, even with a simulator, but you might not encounter the same issue as you were wearing the headset. (P12-VDTD)*

**C7: Lack of Automated Testing.** Developers may face challenges due to the underdeveloped nature of automated testing support in VR. The scarcity of automated or unit tests in VR makes it difficult to minimize manual or repetitive testing tasks. In addition, automated tests are not easy to implement for VR applications by nature. As there is no existing tool in the markets to map the inputs (e.g., user log-in, button presses on the controller, and head movement) in VR to the tests. This phenomenon has been reported by all three groups of participants:

*It's not really easy to automate like testing with scripts. (P2-JD)*

*Even basic things like a 2d traditional UI, testing every button and every combination is a labor-intensive process there. I haven't seen a good way around it. (P8-ED)*

*I think there's certainly more that we could do as engineers in the industry to set up examples of how to apply the tools that exist today to do some automated tests. (P12-VDTD)*

Manual tests are unavoidable in VR testing. However, some of the manual tests can be replaced with automated tests to decrease the labour work and accelerate the development process:

*An ideal version of testing includes, you know, as much automated testing as possible...And when you find something that is actually broken, if you can automate it, you automate it, and write the automation tests for it. And if you can't automate it, you have to actually work with QA to say, Okay, now how do we actually build a proper smoke test to actually go through and have a manual test for this? (P11-VDTD)*

**C8: Inconvenient Collaboration of VR Testing.** Developers might confront obstacles when engaging in collaborative debugging or testing with their colleagues. The issue of collaboration has been explored by Krauß et al. [22]. In their study, the three main challenges faced by collaborative development are: (1) misconceptions about the medium, (2) lack of tool support and (3) missing a common language and shared concepts. Our interviews confirmed and complemented their findings by two aspects in collaborative VR testing: (1) difficult remote debugging and (2) difficult headset sharing.

Remote testing and debugging within a development team remain a critical issue, especially with the adoption of remote work mode in recent years:

*That was a pain by calling and telling the person to change this and that. In order to debug something, I need to tell the person the specific things to do, and the person needs to tell me the result either through screenshot or recording. (P1-JD)*

*When I was helping people debugging, I always cannot see what they saw. It could be ideal to have a mapping between what they do (e.g. click, move) to our aspect. (P4-ED)*

Headset sharing could be an issue since not everyone in the development team has access to the limited number of VR headsets:

*Some people working at home, and do not have VR headsets there. Then they don't have ways to test some VR-specific problems like the performance issues. (P6-ED)*

Even with developers located in the same physical space, collaborative debugging between different developers could be challenging. One of the prominent issues is that the headset sharing between developers need more labour work such as recalibration and communication than developer assumed:

*A teammate head off the headset, then handed me a used VR headset and I put it on, I lost the calibration and was trapped in the box. (P1-JD)*

*Even though I told the other developers what they should do, they started doing other things than you thought. There is a high demand for communication here when you debug with someone else. (P4-ED)*

### 4.3 Comprehensive Challenges

Comprehensive challenges are these challenges beyond the hardware and software limitations. In our study, a lack of standards (C9) and few VR-specific testing support (C10) have been raised. Overcoming these obstacles requires the collective efforts of the entire VR community, as they go beyond the capabilities of individual developers or organizations.

**C9: Lack of Standards.** Developers might grapple with difficulties stemming from an absence of standardized practices. Numerous developers have pointed out the lack of shared standards, such as varying APIs, among diverse VR development tools and software (e.g., Unity and Unreal) as well as hardware (like VR HMDs such as Oculus Quest and HTC Vive).

*I need to use two totally different SDKs for Quest 2 and Vive development, which means I have to double my development work by learning and coding two things. (P6-ED)*

*I think a good driver for this, specifically around standards. I mean, I think of openXR, you know, that's a good industry-inclusive initiative that is trying to get behind alignment for standards, so that everyone follows similar patterns, etc. And they can deploy to as many devices as it is supported within. (P12-VDTD)*

Even though participant feedback from our study suggests that significant progress is still required before developers can fully benefit from the convenience, efficiency, and adaptability that standardization brings to VR development. Organizations like OpenVR strive to address standardization challenges in VR development, which could ultimately benefit many developers:

*I enjoy being able to work on openXR and the really unsexy open standards that are not going to sell front page, you know, news, but ultimately, is really going to benefit developers and the community at large by having open interoperable standards that we as an industry can use. (P12-VDTD)*

**C10: Few VR-specific Testing Support.** Developers might face challenges due to the few VR-specific testing provided by the community and industry. This insufficient support can be evident in multiple forms, such as a limited range of toolkits, tutorials, and documentation, or insufficient cooperation and integration among different tools and solutions:

*I hope to see more samples from the community, and proper documentation, currently they are not straightforward. (P7-JD)*

*Looking for documentation sometimes is still very challenging (P2-JD)*

*Some of the VR development frameworks I use do not have enough information about their technical details. Their internal logic is unknown and unchangeable and I feel like it's a black box. (P6-ED)*

*I'm thinking about the nature of like, VR departments still have a small population compared to trending fields like AI, it doesn't have large community support. (P8-ED)*

However, one of the tool providers found that community support is getting better for VR development:

*There's also the Unity learn portal where there are tutorials and, you know, for all levels, beginning, advanced and professional. So, I believe that there's enough documentation from unity and the tools and packages that we provide to the community that is pretty comprehensive. (P12-VDTD)*

## 5 SURVEY STUDY

To validate and enhance the reliability of our qualitative interview results, we carried out a survey targeting a wider group of VR developers. This approach aimed to triangulate and substantiate our identified challenges.

### 5.1 Study Design

Our survey consisted of three parts. Part 1 contained some demographic questions regarding respondents' years of experience in VR development, the VR development tools they utilized, and the VR headsets they employed. Part 2 was respondents' assessment of each challenge identified during the interview study using a 7-point Likert scale: "Not at all important", "Low importance", "Slightly important", "Neutral", "Moderately important", "Very important", and "Extremely important". To ascertain the validity of the challenges discovered through the interviews, in Part 3 of the survey, we asked respondents to indicate, out of the 10 challenges, the top three challenges they think are most important and relevant to VR testing as well as the top three challenges they think are least important and irrelevant to VR testing.

We recruited participants for our survey through Slack channels of several HCI research communities, VR-related industry communities, as well as a large IT company making VR development software. Additionally, we encourage respondents to share the survey with other VR developers, if feasible. The survey was conducted

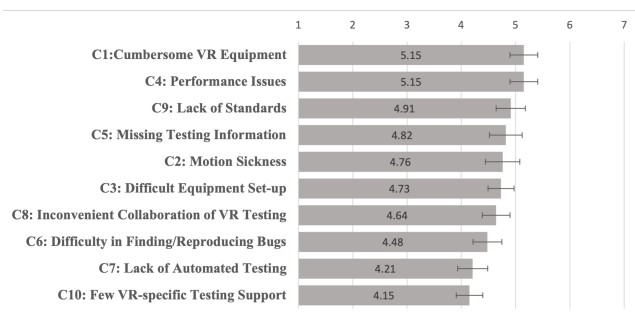

Figure 1: Respondents' ratings on the importance of each challenge on a 7-point Likert scale (1="not at all important", 7="extremely important"; sorted by average ratings).

online via Qualtrics without being supervised by a researcher. No compensation was provided for completing the survey.

## 5.2 Results

A total of 33 VR developers (23 males, 9 females, and 1 non-binary/third gender; aged 23–52) from various organizations participated in our survey, after excluding 4 invalid responses where the respondents do not have enough VR experiences. Among the valid respondents, 16 had 0–2 years of VR development experience, 10 had 3–5 years, and 7 had 6–10 years. All of the participants (33/33) utilized Unity as their VR development tool, while 6 had experience with Unreal, and 1 had used a custom engine for VR development. The most popular headset among respondents was Oculus Quest 1/2, with 26 out of 33 developers have used it. Additionally, 19 respondents had experience with Oculus Rift(s) and 18 with HTC Vive. Furthermore, 13 developers had worked with Google Cardboard, and a few others ($\leq$ 5) had used headsets such as Meta Quest Pro, Varjo XR3, and Valve Index etc. separately.

Figure 1 summarizes the survey responses for the ratings of all identified challenges, including mean values and standard errors. It is evident that all identified challenges have a mean value greater than 4 (neutral), confirming the validity of the challenges. In particular, certain challenges (C1: Cumbersome VR Equipment, C5: Missing Testing Information, and C9: Lack of Standards) exhibit higher mean values, indicating that respondents are more concerned about these issues. C10: Few VR-specific Testing Support has the lowest mean value (4.15). A potential explanation for this could be the recent growth of the VR community due to the popular topic of the Metaverse, which has drawn more developers to the VR community and encouraged organizations and individuals to offer support to developers. C7: Lack of Automated Testing receives the second lowest average rating (4.21). This may be because manual testing cannot be avoided in VR development because of the nature of the software, which requires human labour to check features such as graphics quality and running performance. However, it is also noticeable that 8 of the respondents rated this challenge as "very important" and 1 rated this as "extremely important", which means this challenge still remains prominent among some of the VR developers.

Furthermore, we computed the proportion of each challenge selected as most important and relevant (Figure 2a) as well as that considered least important and irrelevant (Figure 2b). From Figure 2a, we can see that all the challenges have their voters, with C1: Cumbersome VR Equipment (15.15%), C3: Difficult Equipment Set-up (13.13%) and C4: Performance Issues (13.13%) having the most respondents' concern. Notably, we can see that all C1, C3, and C4 are hardware-related challenges, which indicates that the VR development community still has big worries about VR hardware and there is a big growing space for VR hardware. From Figure 2b, we can see that C2: Motion Sickness (13.13%), C4: Performance

Issues (13.13%), and C7: Lack of Automated Testing (13.13%) are the challenges that the respondents are least worried about. Motion sickness, as mentioned by one of the experienced developers in the interviews, could be overcome by the time people develop VR applications and gradually get used to it: *"I am generally immune to motion sickness after spending a lot of time in the VR industry." (P5-ED)*

In summary, all the challenges exhibit reasonable ratings without any outliers, further substantiating the validity of our findings from the interview study.

## 6 FUTURE OPPORTUNITIES

In addition to identifying and verifying the challenges that developers are facing in VR testing, we aimed to explore the opportunities to improve the current state-of-the-art. During the open discussion stage of the interviews, we asked our participants about their ideal VR testing tools or VR testing features. We thus identified several promising avenues that may help with the design of future VR testing tools for both academic and industrial settings.

**Improving hardware design for convenient VR testing:** Hardware-related issues, as pointed out in both interviews and surveys, still remain prominent in VR development communities. Developers suggested some features that could potentially mitigate hardware-related issues in the future. For example, the quick flip on-and-off features would help people switch faster between the VR environment and computer monitor during the testing:

> *I wish there will be a VR headset that I could just wear, flip it off when I need to take a look at my monitor, and flip it back when I need to go back to VR. (P14-VDTD)*

> *What I also really like, is the HoloLens 2 has the visor that can, you know, flip down and flip up... My dream headset, combines this feature in it. (P13-VDTD)*

**Enabling headset-based authoring and testing in VR.** Headset authoring and testing in VR environments were raised a lot during the discussion. First, participants want to see a dedicated debugging mode in VR environments with more flexibility to make some code changes just in VR, without going back to the keyboard:

> *Some functionalities like pre-setting some variables that you will be able to tweak in VR, for example, the width or the height of an object in VR, could save a lot of time of going back and forth. (P3-ED)*

> *I would love to see in headset authoring. So be able to like, put on the headset, and basically have control over your scene hierarchy and inspector. Basically like an engine within the headset, where you're able to edit parameters, move things around (P12-VDTD)*

In addition, participants hoped to see the debugging mode in VR with better testing information visualization:

> *There will be some windows inside the VR to help you see the performance and variables. (P6-ED)*

> *Being able to see different viewpoints, for example, switching between different cameras will help me debug some complicated scenes easily. (P3-ED)*

> *Meta has even some really cool features where you can only have passed through in some areas like you know, the keyboard tracking features are pretty cool, right? I can see my real-world keyboards keep that my real world keyboard so I know where to put my fingers when I am testing in VR (P12-VDTD)*

Generally, participants wanted to have a smooth combination between editing and testing in VR development. One of the tool developers commented:

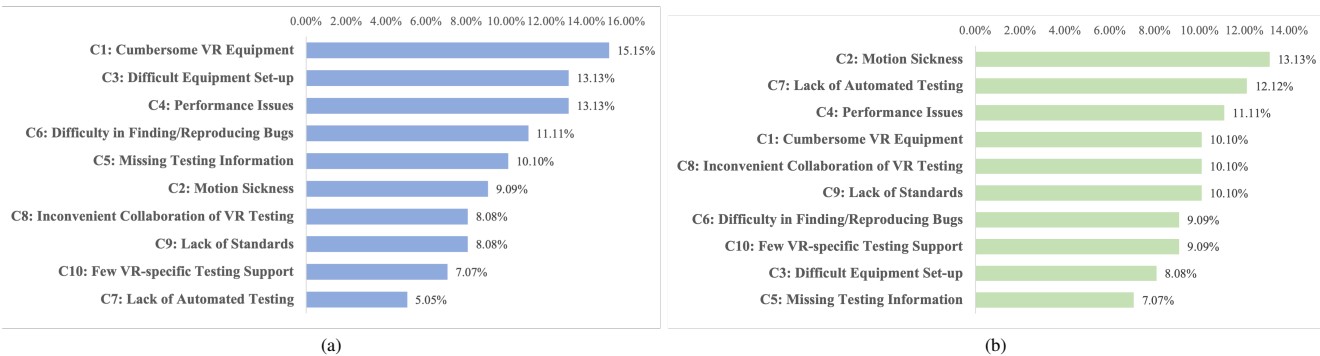

Figure 2: Distribution of the three MOST (a) and LEAST (b) important challenges chosen by respondents (sorted by the number of votes).

*Finally, there should be some unification of editing and testing so that people don't feel a separation in the whole VR development process. (P14-VDTD)*

**Designing collaborative tools for VR testing.** More convenient collaboration in VR development could improve the productivity of the whole VR development team. Communication and collaboration in VR testing can be enhanced with more dedicated VR collaborative tools:

*I do agree that the sort of real-time over a Zoom meeting, and maybe the answer is that Zoom is ultimately not going to be the best platform for real-time VR-related calls. But I think there is sort of a gap there in terms of like, Could we have a zoom alternative in VR where we can load the experience together between developers? (P13-VDTD)*

**Creating automated testing frameworks for VR.** The challenge of automated testing could be mitigated with specialized software frameworks or libraries with functionalities such as button mapping, testing video generation and breaking, playing back, and playing forward some testing timelines.

*It is possible to develop some kind of testing framework, in which you can map some actions from the VR controller to the code to help developers automate some tests. (P9-ED)*

*It is ideal to be able to see the video of the tests and be able to highlight something or catch the model change. (P13-VDTD)*

*It would be helpful to have some systems in place to automate tests and developers could playback or play forward some of the tests in the simulator instead of putting on the VR headset. (P14-VDTD)*

From the hardware perspective, one tool developer also pointed out that other tools like robotic arms could help with automated tests:

*You can have some additional robotic testing, where you can actually say, Okay, what happens when I turn my head this way, what happens when I turn my head this way inside of the headset? (P11-VDTD)*

**Lowering the barrier for junior developers and non-tech users.** On the social level, we could help with the testing difficulties confronted by junior developers by making VR development more accessible, smoothing some steps in the process, and integrating the learning experience into the VR development tools such as Unity engine:

*Most of these headsets are making the VR development not very accessible for a lot of people, whether they have vision*

*disabilities or mobility disabilities, you know, they're very able-bodied devices so far. So I hope that as we move forward, with new devices, and new technologies, all of those things begin to bubble up towards the front and become banners that we carry, and we really, really push forward. (P12-VDTD)*

*Setting up the initial building blocks for a starting project could allow someone without coding knowledge to quickly set up a scene. (P12-VDTD)*

*But also try to make all of the learning happen within the engine so that we can hopefully cut the dependency between a browser-based experience like the learning resource and have that integrated into one of the sample builds. (P12-VDTD)*

Reducing the barriers to entry for VR development could not only draw more developers into the VR community but also significantly contribute to the overall growth and flourishing of the industry.

## 7 LIMITATIONS AND FUTURE WORK

There still exist several limitations in our study that we want to highlight to inform future research. First, as shown in Table 1, a majority of our participants in the interview study uses Unity as the primary VR development tool. This might have biased the challenges we identified, as different development tools may have very different features. Thus a more diverse sample of interviewees could be recruited to further enhance and validate our results.

Second, while our survey study results have confirmed and substantiated the challenges derived from our interview study, more participants could be recruited to better support our insights. However, we do recognize the challenges of getting VR developers as the respondents, not ordinary VR users. Therefore, future studies can be carried out to re-confirm or extend the set of challenges.

Third, we primarily employed a qualitative approach to identify the key challenges for VR testing in our study; while we attempted to use a quantitative method in the survey study to verify the interview results, an even more quantitative way might be appreciated. For example, some logging mechanisms can be implemented in VR development tools to examine the behaviors of VR developers during testing, thus complementing our qualitative results. In deriving the future directions for VR testing tools, prototypes can be built to allow VR developers to actually try different features and then provide their feedback, which could generate richer observations.

Fourth, in our study, we focus on the challenges associated with the testing phase of VR software development; however, there are other challenges that remain prominent in the whole process of development. For instance, the large dependence on game engines such as Unity and Unreal makes VR development especially cumbersome because of the heavy weight of these engines. Also, different versions of game engines and software development tools are updated

almost on a daily basis, which can also be troublesome to the development. These create additional challenges for VR testing that are associated with the software-related and comprehensive challenges we identified (especially C8, C9, and C10). Thus, future investigations regarding the whole development process of VR software can be carried out to nail down other challenges and opportunities that developers may face.

In summary, several future efforts should be made to continue this line of research, and our study has opened doors to a wide range of development and investigation opportunities for VR testing.

## 8 CONCLUSION

In this paper, we aimed to fill in the gap in the literature by exploring the challenges, needs, and future opportunities of software testing in VR development. During the interviews, participants expressed various difficulties encountered during the VR development testing phase, which we then consolidated into a list of challenges. Moreover, we explored future directions for VR testing and presented the outcomes that may shed light on the research and technology development. We confirmed the challenges through a survey, analyzing the ratings given for each challenge. Our findings highlight multiple design opportunities for both academic and industrial stakeholders to alleviate these VR testing challenges. By addressing these issues and following our results, we believe that future development can enhance the utility, productivity, and overall development experience for VR developers during the testing phase.

### ACKNOWLEDGMENTS

We wish to thank all the participants for their effort and feedback. We especially thank Alan Liwei Wu (University of Waterloo) as well as Valerio Ortenzi and Monika Underwood (Unity Technologies) for their valuable input on this project. This research is supported in part by the Mitacs Accelerate program via collaboration with Unity Technologies.

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
