# OpenReview forum: "Challenges and Opportunities for Software Testing in Virtual Reality Application Development"
_graphicsinterface.org/Graphics_Interface/2023/Conference_SD — GI 2023 - second deadline_

### Official Review · Reviewer_wbH8 · 2023-04-21
**A good contribution to the VR/AR Community**

**Rating:** 7
**Confidence:** 3

**Review:**

Software testing for the development of VR applications has challenges and has not been explored much. To address this, the authors conducted a qualitative interview study with 14 professional VR developers and validated the findings of the qualitative study with a second survey study with 33 new participants.

The paper is very well-written and has a strong literature review. The two studies are well thought out and the results, backed up by specific participant responses, are presented in thorough detail. Also, the summary of the results presented in tabular format is helpful for readers who prefer to quickly scan the paper. Having decent experience with VR development myself, I do agree with their findings and thus believe that the results of the first study are valid. More importantly, the results of the first study are validated by the second survey study, making this work a good contribution to the VR/AR community. Having said that, I do have some suggestions for the authors to further improve their work.

It would be good to see if the authors could extend their discussion on the challenges associated with collaboration during the development phase as VR software is generally quite heavyweight. Also, different versions of Unity/Unreal are coming out almost every day which can also be devastating to the development. I am curious if the authors investigated this as well.

Were the interviews conducted in person or virtually? Similarly, did the participants filled the survey online? Were the surveys supervised by a researcher?

#### MINOR ISSUES
There are some minor typos in the paper. For example, a dot is missing in Section 5.1. There are other places with similar problems, e.g., a typo in the caption of Fig. 2.

---

### Official Review · Reviewer_9zWx · 2023-04-23
**From the expert interview, the authors derived ten key challenges in the domain of VR testing, which they then validated in the second survey.**

**Rating:** 6
**Confidence:** 3

**Review:**

This work comprises two empirical studies and corresponding analyses: an expert interview with 14 VR practitioners and an extended survey with 33 participants of varying VR experience levels. From the expert interview, the authors derived ten key challenges in the domain of VR testing, which they then validated in the second survey.


This topic is timely, and the detailed analysis of the expert review proposed some interesting directions to improve, such as the difficulties in reproducing bugs and the lack of automated testing. While the analysis of the second study is somewhat weak, mostly consisting of a comparison of the numerical results of the Likert scale over the ten challenges they proposed. This part can be improved to provides more valuable insights.


The reading experience was smooth, except for Section 2.1 (which lists general types of VR applications). It is not immediately clear how these contribute to identifying the challenges in VR testing activities.
I suggest condensing these paragraphs to briefly highlight the motivation, and perhaps moving & merging them to the introduction rather than staying as a standalone subsection in the related work.


Based on the quality of the manuscript, I am leaning towards weak acceptance.

Minor: Sec 4.2-C8-3rd paragraph has some broken quotes ("I need to the person" and "you the specific things to do"), which seem like typo during audio transcription. (But please indicate if those are the speaker's original words, then it's fine)

Minor: Fig. 2 caption "LEAT" to "LEAST".

---

### Official Review · Reviewer_XesX · 2023-04-25
**Solid paper about a real-world challenge that needs addressing**

**Rating:** 8
**Confidence:** 4

**Review:**

This is a well written, interesting piece of research, tackling an identified real-world challenge of increasing importance and relevance to the HCI community.
The problem is well motivated and the potential impact of the work is good.
The method employed to look into the challenges in VR testing is appropriate and relatively straightforward (although it is not mentioned whether the interviews and the surveys were administered in person or online). The data analysis makes sense for the collected data as well. The results from the first study are very interesting. The second study is a bit more anecdotal. Although it is great to see the authors confirm the findings from the first study, I  think that this section could be condensed even more. In particular, the sentences at the end of this section that repeat what the figure shows are not very useful. one thing the authors might want to consider is to sort the bars in descending order in Figure 1. Same with Figure 2; it might be interesting to sort each subfigure, and then to connect each bar (challenge) to its corresponding bar (challenge) in the other figure.

The one aspect of the paper that I find confusing is the use of quotes, i.e. providing new data, in the discussion. A discussion should provide a perspective on the results, not add new results, so it makes the whole results/discussion blurry. Why were these quotes not introduced in the results? Why is the future opportunities section not a part of the results?

Other minor comments:

I do not find Section 2.1 to be much needed/relavant to the research problem at hand; VR has been around for a long time enough to not need to be justified as a mature (often applied) research area. It is a good candidate for cutting, which will make the paper more focused.

I am not familiar with all the technologies presented in Section 2, but I do appreciate the detailed list of approaches, tools and technologies for VR development that readers can benefit from.

The recruitment method could be better detailed, beyond "reaching out" to potential participants. Also, if ethics approval was obtained, please mention it.

Typo in caption of Figure 2: LEAT -> LEAST